# Silencing of lncRNA *AK045490* Promotes Osteoblast Differentiation and Bone Formation via β-Catenin/TCF1/Runx2 Signaling Axis

**DOI:** 10.3390/ijms20246229

**Published:** 2019-12-10

**Authors:** Dijie Li, Ye Tian, Chong Yin, Ying Huai, Yipu Zhao, Peihong Su, Xue Wang, Jiawei Pei, Kewen Zhang, Chaofei Yang, Kai Dang, Shanfeng Jiang, Zhiping Miao, Meng Li, Qiang Hao, Ge Zhang, Airong Qian

**Affiliations:** 1Lab for Bone Metabolism, Key Lab for Space Biosciences and Biotechnology, School of Life Sciences, Northwestern Polytechnical University, Xi’an 710072, China; lidijie@mail.nwpu.edu.cn (D.L.); tianye@nwpu.edu.cn (Y.T.); yinchong42@nwpu.edu.cn (C.Y.); huai_@mail.nwpu.edu.cn (Y.H.); zhaoyipu@mail.nwpu.edu.cn (Y.Z.); suph@mail.nwpu.edu.cn (P.S.); wangxue1005@nwpu.edu.cn (X.W.); peijiawei@mail.nwpu.edu.cn (J.P.); zkw@mail.nwpu.edu.cn (K.Z.); yang2015@mail.nwpu.edu.cn (C.Y.); dangkaipower@hotmail.com (K.D.); miaozp@nwpu.edu.cn (Z.M.); 2Research Center for Special Medicine and Health Systems Engineering, School of Life Sciences, Northwestern Polytechnical University, Xi’an 710072, China; 3NPU-UAB Joint Laboratory for Bone Metabolism, School of Life Sciences, Northwestern Polytechnical University, Xi’an 710072, China; 4Law Sau Fai Institute for Advancing Translational Medicine in Bone and Joint Diseases, School of Chinese Medicine, Hong Kong Baptist University, Hong Kong 999077, SAR, China; 5State Key Laboratory of Cancer Biology, Biotechnology Center, School of Pharmacy, Fourth Military Medical University, Xi’an 710032, China; limeng@fmmu.edu.cn (M.L.); haosuq@fmmu.edu.cn (Q.H.); 6Institute of Integrated Bioinfomedicine and Translational Science, School of Chinese Medicine, Hong Kong Baptist University, Hong Kong 999077, SAR, China

**Keywords:** *AK045490*, lncRNA, osteoblast differentiation, bone formation, osteoporosis, transcript factor

## Abstract

Osteoporosis, a disease characterized by both loss of bone mass and structural deterioration of bone, is the most common reason for a broken bone among the elderly. It is known that the attenuated differentiation ability of osteogenic cells has been regarded as one of the greatest contributors to age-related bone formation reduction. However, the effects of current therapies are still unsatisfactory. In this study we identify a novel long noncoding RNA *AK045490* which is correlated with osteogenic differentiation and enriched in skeletal tissues of mice. In vitro analysis of bone-derived mesenchymal stem cells (BMSCs) showed that *AK045490* inhibited osteoblast differentiation. In vivo inhibition of *AK045490* by its small interfering RNA rescued bone formation in ovariectomized osteoporosis mice model. Mechanistically, *AK045490* inhibited the nuclear translocation of β-catenin and downregulated the expression of TCF1, LEF1, and Runx2. The results suggest that Lnc-*AK045490* suppresses β-catenin/TCF1/Runx2 signaling and inhibits osteoblast differentiation and bone formation, providing a novel mechanism of osteogenic differentiation and a potential drug target for osteoporosis.

## 1. Introduction

Osteoporosis is an age-related bone disease and it can be defined as the deterioration in bone mass and micro-architecture, along with increasing risk to fragility fractures. Age is recognized as the major risk factor of osteoporosis [1]. Postmenopausal osteoporosis, as a result of estrogen deficiency, is the most common type of osteoporosis [2]. It is known that the attenuated differentiation ability of osteogenic cells has been regarded as one of the greatest contributors to age-related bone formation reduction. The osteoblast differentiation is precisely controlled by a variety of intracellular signaling pathways. In recent years, emerging evidence has demonstrated that epigenetic pathways, including long noncoding RNAs (lncRNAs) [3], are playing critical roles in regulating osteoblast differentiation and thus considered as a new focus point in bone research.

LncRNAs, one kind of noncoding RNA with more than 200 nucleotides in length, are found to be widely expressed in eukaryotes and playing various roles in multiple physiological and pathological processes [4]. Several lncRNAs have been identified to be essential in regulating osteoblastic differentiation of stem cells [5,6]. For example, LncRNA *AK045490* inhibits osteoblast differentiation and bone formation by regulating transcription factor T cell factor 1(TCF1)/lymphoid enhancer-binding factor 1(LEF1) activity in mouse mesenchymal stem cells (mMSCs) [7]. Linc-ROR promotes osteogenic differentiation of human bone-marrow-derived mesenchymal stem cells (hMSCs) via activating Wnt/β-catenin pathway [8]. These studies suggest that it is desirable to make further investigation of the lncRNAs on the aspect of regulating osteoblast differentiation.

In this study, we revealed that lncRNA *AK045490* was negatively associated with osteoblast differentiation and bone formation. In vitro knockdown of *AK045490* could promote β-catenin nuclear translocation and up-regulates the expression of TCF1, LEF1, and Runt-related transcription factor 2 (Runx2). The molecular mechanism of *AK045490* in inhibiting osteogenesis was also investigated by evaluating the expression and activities of osteogenic transcription factors. Finally, the ovariectomized (OVX) mice were used to clarify the promoting effect of *AK045490* siRNA on bone formation in postmenopausal osteoporosis.

## 2. Results

### 2.1. Elevated AK045490 Expression in Bone Was Accompanied by Deteriorated Bone Microstructure and Decreased Bone Formation in Osteoporotic Mice

In our previous study, we have screened osteogenic lncRNAs through mRNA/lncRNA microarray combined with gene co-expression analysis. We speculate that *AK045490* might be one of the osteoblastic differentiation inhibiting lncRNAs [7]. To determine the expression level of *AK045490*, we established two osteoporotic mice models, age-related osteoporotic mice model (aging mice) and postmenopausal osteoporotic mice model (ovariectomized mice, OVX mice). In male aging mice, we found that the expression of AK045490 in the aged group was 20 times more than that in the young group (Figure 1a). Meanwhile, both the bone mineral density (BMD) and mineral apposition rate (MAR) were lower in the aged mice, when compared to the young (Figure 1b). Similarly, in female mice, the expression of *AK045490* in the OVX group was significantly higher in the OVX group, when compared to the sham-operated (Sham) group (Figure 1c). The BMD and MAR were lower in the OVX mice, when compared to the Sham group (Figure 1d). The above results suggested that the decreased bone formation and the weakened bone microstructure are accompanied by increased *AK045490* expression level.

### 2.2. AK045490 Inhibited Osteoblast Differentiation

To investigate the role of *AK045490* in osteoblast differentiation, MC3T3-E1 cells were treated with *AK045490* siRNA (si-*AK045490*) or negative control siRNA (si-NC), respectively. The expression level of *AK045490* in MC3T3-E1 cells was decreased by 62% after *AK045490* siRNA transfection, when compared to negative control (Figure 2a). In the *AK045490* siRNA transfection group, mRNA expression levels of osteogenic marker genes, bone specific alkaline phosphatase (*Alp*), Osteocalcin (*Ocn*) and collagen type I (*Col Iα1*) were considerably upregulated by 76%, 21%, and 34%, respectively, when compared to the si-NC group (Figure 2b). In addition, the alkaline phosphatase activity, which was detected by ALP-positive blue-violet complexes staining, was significantly increased in the *AK045490* siRNA group (Figure 2c, up). The number of mineralized nodules, which was detected by Alizarin Red-staining, was increased in the *AK045490* siRNA group as well (Figure 2c, bottom). The above results suggested that *AK045490* played a negative role in osteoblast differentiation.

### 2.3. Knockdown of AK045490 Promoted β-Catenin Nuclear Translocation and Up-Regulates the Expression of TCF1, LEF1 and Runx2

As β-catenin/TCF signaling is reported to play a key regulatory role in osteoblast differentiation by targeting Runx2 [9] and the Runx2 expression was up-regulated by *AK045490* knockdown as shown below, we took efforts to study whether knockdown of *AK045490* would enhance β-catenin signaling, and thereby promoting osteoblast differentiation. As the nuclear translocation of β-catenin is the hallmark of β-catenin signaling activation, we examined the cellular localization and levels of β-catenin by both immunohistofluorescence (IHF) staining and western blot (WB) analyses. IHF staining showed that, in the control cells, the β-catenin was mainly distributed around the cytoplasm and partially in the nucleus (Figure 3a, up). In the *AK045490* knockdown cells (transfected with *AK045490* siRNA), β-catenin was mainly located at cell nucleus and there was less β-catenin staining around the cytoplasm (Figure 3a, bottom). This result suggested that knockdown of *AK045490* promoted the nuclear translocation of β-catenin. Further, analyses of the cytosolic and nuclear fractions of cellular proteins by western blot also showed that there was a significant increment of the nuclear amount of β-catenin in *AK045490* knockdown cells (Figure 3b,c). Furthermore, there was a 45% increase in the transcription activity of TCF1 in *AK045490* knockdown cells, when compared to that in control cells (Figure 3e). Consistently, gene expression levels of both *Tcf1*, *Lef1*, and *Runx2* were significantly increased in the *AK045490* knockdown cells compared to control cells (Figure 3d). As β-catenin/TCF1 directly targets Runx2, all the above results indicated that *AK045490* suppressed β-catenin signaling in osteoblasts. These results also suggested that *AK045490* might negatively regulate osteoblast differentiation via the β-catenin/TCF1/Runx2 pathway. In addition, we speculated that AK045490 might function as competing endogenous RNA (ceRNA), leading to the liberation of corresponding miRNA targeted transcripts. To test this hypothesis for further investigation, the lncRNA–microRNA–mRNA interaction was predicted by bioinformation analysis. It was found that *AK045490* might be an upstream regulator of the microRNAs that potentially target bone-associated signaling pathway, such as miR-6344, miR-3089, and miR-6951 (Figure 3f).

### 2.4. Promoting Effect of AK045490 siRNA on Calvaria Bone Formation in OVX Mice

To investigate whether interference of *AK045490* expression in vivo could rescue the age-related reduction of bone formation, we established ovariectomized (OVX) mouse model. C57BL/6 mice were used to simulate the postmenopausal osteoporosis, a common aging-related bone disease. The *AK045490* siRNA was transfected every other day in calvaria of OVX mice (twice per day) with negative control siRNA (si-NC) as control group. It was showed that the mineral apposition rate (MAR; a parameter for assessment of bone formation) in the OVX group was markedly decreased compared to that in Sham group and increased significantly after *AK045490* siRNA treatment compared to si-NC group (Figure 4a,b), suggesting that there was a promoting effect of *AK045490* siRNA on calvarial bone formation in OVX mice.

## 3. Discussion

Age-related osteoporosis is associated with an increase of remodeling and a negative remodeling balance in individual bone remodeling units [10]. Postmenopausal osteoporosis, as a result of estrogen deficiency, is the most common type of osteoporosis and leads to osteoporotic fractures. There is a large variation in reported incidence of fractures, but an average up to 50% of women above 50 years old are at risk of fractures [2]. However, the current counter-measures on osteoporosis have either side effects or poor patient compliance. So, it is desirable to develop novel therapeutic methods for anti-osteoporosis.

LncRNA have been widely investigated in a wide variety of physiological and pathological processes. Emerging evidences show that lncRNAs play important roles in regulating cell differentiation. To date, many lncRNAs have been considered to regulate osteogenic differentiation [11]. Using human whole transcriptome microarray, Zhang et al. profiled the expression and analyzed the functional network of lncRNAs during osteogenic differentiation of human bone marrow mesenchymal stem cells (hBMSC), in which a total of 1408 lncRNAs with 623 lncRNAs downregulating and 785 lncRNAs significantly upregulating were detected along with osteogenic differentiation [12]. In this study, we screened osteogenic-related lncRNAs in our established preosteoblast cell line [13] by performing mRNA and lncRNA microarray in combination with KEGG analysis and PCC co-expression analysis [7]. Among them, AK045490 was one of lncRNAs that showed the most significant increment [7]. We further identified that AK045490 was highly expressed in two osteoporotic mice model, age-related osteoporotic mice and postmenopausal osteoporotic mice (Figure 1), indicating its negative effect on osteogenic differentiation.

We further confirmed the regulatory functions of AK045490 in osteoblast differentiation. In MC3T3-E1 cells, knockdown of AK045490 significantly increased the expression of osteoblast differentiation marker genes (*Alp*, *Ocn*, and *Col Iα1*), the ALP activity and the formation of mineralized nodules (Figure 2), which were consistent with the phenomenon that the elevated AK045490 expression in bone tissue was accompanied by deteriorated bone microstructure and decreased bone formation in aging mice, as well as in OVX mice. Several canonical signaling pathways, such as the TGF-beta signaling pathway [14,15,16], the MAPK signaling pathway [17], the Hippo signaling pathway [18], the Notch signaling pathway [19], the Jak-STAT signaling pathway, and the Toll-like receptor signaling pathway [12], which are associated with the osteoblast differentiation, have been found to be closely related to lncRNAs. All of these findings indicated that lncRNAs are critical elements in osteoblast differentiation. It has been widely recognized that β-catenin/TCF signaling plays a key role in regulating osteoblast differentiation by targeting Runx2 [9]. Moreover, our previous study demonstrated that MACF1 play a positive regulatory role in osteoblast differentiation and bone formation by promoting nuclear translocation of β-catenin/TCF1/Runx2 signaling axis [20,21]. RUNX2, TCF1, and LEF1 were master transcription factors regulating osteoblast differentiation [22,23]. We found that knockdown of AK045490 could promote the nuclear translocation of β-catenin and up-regulate the expression of transcription factor TCF1, LEF1, and Runx2 (Figure 3a–e), indicting a potential pathway that AK045490 suppressed osteoblast differentiation and bone formation. In the predicted network of lncRNA–microRNA–mRNA interaction, we found that AK045490 might be an upstream regulator of several microRNAs, which potentially target bone-associated signaling pathway (Figure 3f), such as miR-6344 targeting KREMEN1 (an antagonistic modulator of the Wnt signaling pathway, involving osteogenesis [24,25]), and miR-3089 targeting GSK-3 beta (an negative controller of the Akt-GSK3 beta-NFATc1 signaling pathway, involving osteoclastogenesis [26,27]). It might be a possible direction to understand the in-depth molecular mechanism of AK045490 in regulating osteoblast differentiation.

Due to the importance of lncRNAs in regulating osteogenic differentiation, more and more researchers have started to explore RNA based therapy directing to lncRNAs in bone metabolic disorders. For example, by using H19-overexpressed human bone-marrow mesenchymal stem cells (hBMSCs) in mice, Huang et al. investigated the role of lncRNA H19 in heterotopic bone formation [14]. By establishing depleted and transgenic overexpression of lnc-Bmncr mice, Li et al. evaluated the function of Bmncr in BMSCs on age-related switch between osteoblast and adipocyte differentiation [28]. It was proved that intravenously administered siHOXC-AS3 was effective in prevention of bone loss, sustained by both anticatabolic activities and bone-forming in mouse models [29]. Silencing lncRNA MEG3 in tibia fracture mice models by siRNA could accelerate tibia fraction healing via activating the Wnt/beta-catenin signaling pathway [30]. In the present study, we used MC3T3 cell line to investigate the molecular mechanism. It had better to use primary osteoblasts instead of transformed cells, which has some drawbacks, such as that it might be regulated in different ways. In addition, cell proliferation is also an important contributor to osteoblastogenesis, which is difficult to study with transformed cells. Recently, Mulati et al. reported that the lncRNA Crnde affects mainly osteoblastogenesis through proliferation and thus bone formation, also via Wnt/β-catenin signaling [31]. So far, there are only a few rescue studies being reported about utilizing lncRNAs siRNAs in systemic osteoporosis animal model. In the current study, we investigated the role of AK045490 in the calvarial formation using OVX mice, a postmenopausal osteoporotic mouse model. We found that the mineral apposition rate was noticeably higher in AK045490 siRNA treatment group compared to Negative Control siRNA treatment group (Figure 4), indicating that inhibition of AK045492 could relieve the bone formation reduction in postmenopausal osteoporosis. This study provided an evidence that interference of lncRNA by siRNA in vivo would recover the reduction of bone formation in ovariectomized mice. To date, AK045490 in humans has not been reported. Our next step will involve investigating AK045490 in human osteoblast-like cell line. Our data substantiated that inhibition of suppressive lncRNA of osteogenesis in osteoblasts might be a potential anabolic strategy for ameliorating the reduction of bone formation in postmenopausal osteoporosis.

In summary, in this study we identified lncRNA AK045490 as an inhibitor for osteoblast differentiation and bone formation. In the case of mechanism, AK045490 might suppress osteoblast differentiation through inhibiting β-catenin/TCF1/Runx2 signaling. In addition, knockdown of AK045492 could relieve the bone formation reduction in ovariectomized osteoporotic mice. These findings might reveal a novel mechanism of osteogenic differentiation and provide a potential drug target for osteoporosis.

## 4. Materials and Methods

### 4.1. Cell Culture

The MC3T3-E1 clone 14 cell line was maintained in Alpha Modified Eagle’s Medium (α-MEM, Gibco, Carlsbad, CA, USA) containing 10% FBS (Gibco) and 1% penicillin and streptomycin (Gibco). The cells were maintained under standard cell culture conditions of 5% CO_2_ and 95% humidity and were not used beyond passage 25. For the experiments, confluent cells were detached using 0.25% trypsin containing 10 mM EDTA, resuspended in antibiotic-free growth medium and plated onto six-well plates at a density of 200,000 cells per well.

### 4.2. Mice Model

Aging and ovariectomized (OVX) mice were adopted to construct the osteoporosis model. All mice were purchased from the Laboratory Animal Center of the Fourth Military Medical University (Xi’an, China). For aging mice model, six 6-month-old and six 18-month-old male C57BL/6 mice were maintained under standard animal housing conditions (12 h light and 12 h dark cycles and free access to food and water). Mice were euthanized and femurs were collected and processed for microCT analysis, bone histomorphometry analysis and bone-derived MSCs isolation (*n* = 6/group). For OVX mouse model, 2-month-old female C57BL/6 mice were maintained under standard animal housing conditions. The mice were ovariectomized or sham-operated at 3 months of age. Mice were euthanized 5 weeks after surgery (4 months of age) and calvarias were collected. All animal experiments were performed in accordance with the recommendation of “the Guiding Principles for the Care and Use of Laboratory Animals” (the Institutional Experimental Animal Committee of Northwestern Polytechnical University, Xi’an, China) and the trial was approved by the Institutional Experimental Animal Committee of Northwestern Polytechnical University, Xi’an, China (7 Mar 2019) and was registered (No.2019030, 7 Mar 2019). For all procedures involving animals, all efforts were made to reduce the number of the mice used and their suffering.

### 4.3. RNA Isolation and Real-Time PCR (RT-PCR) Analysis

Total RNA was extracted from frozen tissue with a polytron homogenizer (Prima PB100, Shanghai, China) or directly from cells by using TRIzol reagent (Lefe technologies^TM^, Carlsbad, CA, USA). Total RNA was used as a template for double-stranded cDNA synthesis (PrimeScript^TM^ RT reagent Kit, Takara, Dalian, China). The SYBR^®^ Premix Ex Taq^TM^ II (Takara, Dalian, China) was applied for the quantitative RT-PCR. GAPDH was used as endogenous controls for normalization. All the primer sequences designed and used are listed in Table 1. The relative fold changes of candidate genes were analyzed by using 2^−ΔΔCT^ method [32].

### 4.4. MicroCT Analysis

The femur was scanned by the microCT system (v. 6.5, viva CT40, SCANCO Medical, Bruettisellen, Switzerland) and the left distal femoral metaphysis was analyzed. Briefly, the femur was dissected from the mice free of soft tissue, fixed overnight in 70% ethanol and analyzed by microCT. Images of femurs were reconstructed and calibrated at the isotropic voxel size of 10.5 µm, respectively (70 kVp, 114 µA, 200 ms integration time, 260 thresholds, 1200 mg HA/cm^3^). Using the Scanco evaluation software, regions of interest (ROIs) were defined for trabecular parameters. For the left distal femoral metaphysis, the entire femora were reoriented with the mid-diaphysis parallel to the z-axis, and bone length was measured as the distance between the most proximal and distal transverse plans containing the femur. Starting from the most proximal aspect of the growth plate, the trabeculae region on 100 consecutive slices were selected. The trabeculae were analyzed by manually contouring excluding the cortical bone for three-dimensional reconstruction (sigma = 1.2, supports = 2 and threshold = 200) to calculate the following trabeculae parameters: Tb.BV/TV (trabecular bone volume per total volume), Tb.vBMD (trabecular volumetric bone mineral density).

### 4.5. Bone Histology and Histomorphometry

The distal femur was fixed in 4% paraformaldehyde (PFA), dehydrated in graded concentrations of sucrose and embedded without decalcification in the optimal cutting temperature compound (OCT) (14020108926; Leica, Wetzlar, Germany) using our previously established protocol [7]. After dehydration, frontal sections for trabecular bone were obtained from the distal femur at a thickness of 5 µm with Leica SM2500E microtome (Leica Microsystems). Fluorescence micrographs for the calcein in the bone sections were captured by a fluorescence microscope (Leica image analysis system). Bone dynamic histomorphometric analyses for MAR (mineralizing apposition rate).

### 4.6. siRNA Transfection In Vitro

Transient transfection of cells with siRNA was performed in 24-well plates using Lipofectamine 3000 reagent (Lefe technologies^TM^, Carlsbad, CA, USA). si-NC (50 nM) or siRNA (50 nM) was transfected into cells respectively in culture medium and then harvested for further detection. AK045490 siRNA sequences designed and used in this case are: 5′-GCAGCUGUGCAGUCAGAUUdTdT-3′ (sense strand). 5′-AAUCUGACUGCACAGCUGCdTdT-3′ (antisense strand).

### 4.7. Western Blot

Cells were lysed in a buffer containing 50 mM Tris (pH 7.5), 150 mM NaCl, 0.5% NP40 (Beyotime, Beijing, China), and protease inhibitors (TaKaRa, Dalian, China). Proteins in lysates were resolved by SDS–PAGE and then transferred to polyvinylidene difluoride membranes. The membranes were blocked with 5% BSA for 1 h at room temperature and incubated with primary antibody recognizing β-catenin (Cell Signaling Technology, Danvers, MA, USA), Lamin B (Cell Signaling Technology, Danvers, MA, USA), or Gapdh (Boster, BM1623, Wuhan, China) at 4 °C overnight. Incubation with secondary horseradish peroxidase-labelled antibody was carried out for 1 h at room temperature. Protein bands were visualized by chemiluminescence using an Electro-Chemi-Luminescence (ECL) kit (Proteintech, Hubei, China) and exposed to X-ray film. Protein band intensities were quantified using Image-J software v. 1.45 (National Institutes of Health, NIH, Bethesda, MD, USA). The GAPDH was used as the internal control for total protein and cytoplasmic protein, and Lamin B1 was used as the internal control for nuclear protein.

### 4.8. ALP Staining

Alkaline phosphatase staining was monitored using an Alkaline Phosphatase Assay Kit (C3206, Beyotime, Beijing, China). Cells were fixed by immersion 4% PFA solution for 10 min and rinsed in PBS for 5 min 3 times. The samples were then placed in an alkaline phosphatase staining solution for 30 min. The whole procedure was protected from light. After discarding the solution, cells were rinsed in deionized water 2 min 2 times.

### 4.9. Alizarin Red Staining

Cells were fixed by immersion 4% PFA solution for 10 min and rinsed in PBS for 5 min 3 times. Next, the nodules were stained with 0.5% Alizarin Red S (A5533, Sigma-Aldrich, St. Louis, MO, USA), pH 4.0, for 20 min with gentle agitation. Cells were rinsed five times with double-distilled H_2_O. The plates were dried and scanned with CanoScan 9000F Mark II scanner.

### 4.10. Immunohistofluorescence (IHF) Staining

Cells were fixed with 4% PFA for 20 min at room temperature followed by permeabilization with 0.2% Triton X-100 for 2 min. Next, samples were blocked with 1% BSA diluted in 1 × PBS supplemented with 0.05% Tween-20 for 45 min and sequentially incubated with primary antibody recognizing β-catenin (1:50, Cell Signaling Technology, Danvers, MA, USA) at 4 °C overnight and fluorescently labeled secondary antibodies for 1 h. After washing with PBST (0.05% Tween-20), cells were incubated with PE-conjugated goat anti-rabbit IgG secondary antibody (1:100, Hangzhou HuaAn Biotechnology, Hangzhou, China) for 90 min at room temperature. 4′,6-Diamidino-2-phenylindole (DAPI, 1 µg/mL, Beyotime, Beijing, China) was used to counterstain cell nuclei for 5 min at room temperature.

### 4.11. siRNA Transfection In Vivo

Thirty-two male C57BL/6 mice were randomly divided into six groups (si-NC, and si-AK045490). The transfection was performed once/3 days, for 2 times after ovariectomy operation (OVX), respectively. In the si-AK045490 group, mice were injected subcutaneously over the calvarial surface with AK045490 siRNA formulated with Entranster™ In Vivo Transfection Reagent (18668-11-2; Engreen Biosystem Co., Ltd., Auckland, New Zealand) at the dosage of 40 µL according to the manufacturer’s instructions. In the si-NC group, mice were injected with negative control siRNA (si-NC) mixed with the same transfection reagent. In the Mock group, the calvarial surface of mice received the same volume of normal saline mixed with transfection reagent. In the baseline (BL) group and OVX group, mice were given no treatment. All mice received the same standard diet during the experimental period. Mice were euthanized 37 days after OVX and calvarias were collected. For baseline group, mice were euthanized 10 days after OVX treatment.

### 4.12. Analysis of lncRNA–microRNA–mRNA Interaction

First, the microRNAs associated with AK045490.1 were predicted from miRDB Database (http://www.mirdb.org/, 5 September 2019). The top 10 microRNAs, which have a correlation score greater than 90, were considered as the candidates. Second, the target genes of the 10 microRNA candidates were predicted from miRWalk Database v. 3.0 (http://mirwalk.umm.uni-heidelberg.de/, 5 September 2019). Third, the target genes of the above microRNAs were obtained by intersecting the miRNA target genes with the osteogenic differentiation related genes predicted in Genecard database (https://www.genecards.org/, 15 September 2019). Finally, the LncRNA–microRNA–RNA interaction network was constructed by using Cytoscape (http://cytoscapeweb.cytoscape.org/, 22 September 2019).

### 4.13. Statistical Analyses

All experiments were independently repeated at least three times with each performed in triplicate. Statistical analyses of the data were performed using the GraphPad Prism v. 6 software (GraphPad Software, La Jolla, CA, USA) and the Student *t*-test was used. All data were reported as the mean ± SD and *p* < 0.05 were considered statistically significant for all comparisons.

## Figures and Tables

**Figure 1 ijms-20-06229-f001:**
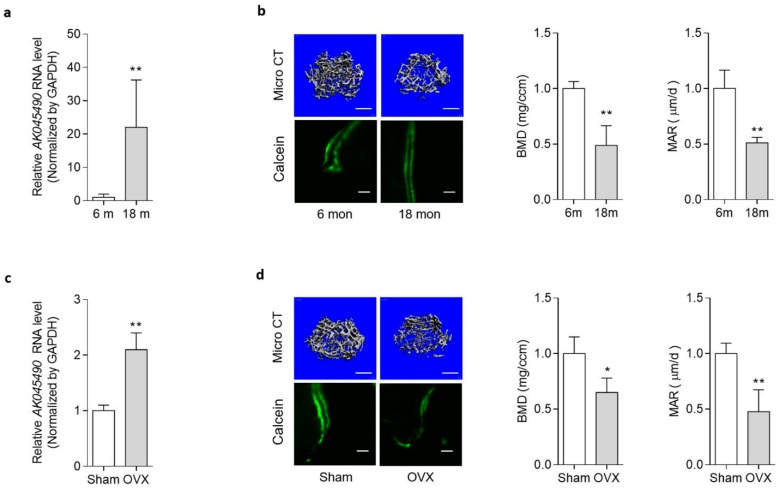
Elevated *AK045490* expression in bone is accompanied by deteriorated bone microstructure and decreased bone formation in aging mice and in ovariectomized (OVX) mice. (**a**) The RNA level of long noncoding RNAs (lncRNAs) *AK045490* in bone isolated from the age-related osteoporotic mice. (**b**) Representative images showing the 3D architecture (Left, top) and Micro Computed Tomography (Micro CT) measurements in the distal femurs (Middle). Representative images of new bone formation assessed by double calcein labeling (Left, bottom) and quantitative analysis of mineral apposition rate (MAR) at the distal femur (Right). (**c**) The RNA level of lncRNA *AK045490* in bone isolated from the postmenopausal osteoporotic mice. Sham: Sham operation group. OVX: ovariectomy operation group. (**d**) Representative images showing the 3D architecture (Left, top) and Micro CT measurements in the distal femurs (Middle). Representative images of new bone formation assessed by double calcein labeling (Left, bottom) and quantitative analysis of mineral apposition rate (MAR) at the distal femur (Right). All data were expressed as mean ± SD. Student’s *t*-test was performed for comparison between two groups. *p* value less than 0.05 were considered significant in all cases (* *p* < 0.05, ** *p* < 0.01). Scale bar: 500 μm in b, d (top), 20 μm in b, d (bottom). *n* = 6 mice in each group.

**Figure 2 ijms-20-06229-f002:**
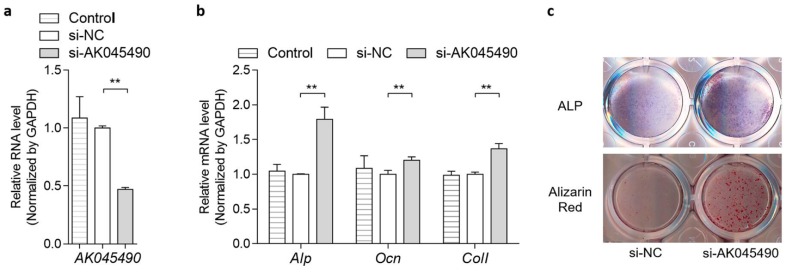
Silencing of *AK045490* promoted osteoblast differentiation. (**a**) AK045490 RNA levels of MC3T3-E1 cells treated with *AK045490* siRNA, negative control RNA (si-NC) or without treatment (Control), as detected by RT-PCR. (**b**) *Alp, Ocn, and Col-**I* expression levels of MC3T3-E1 cells treated with *AK045490* siRNA, as detected by RT-PCR. (**c**) ALP staining (up) and Alizarin Red staining (bottom) in MC3T3-E1 cells treated with *AK045490* siRNA. All data were expressed as mean ± SD. Student’s *t*-test was performed for comparison between two groups. *p* values less than 0.05 were considered significant in all cases (** *p* < 0.01). *n* = 3 in each group.

**Figure 3 ijms-20-06229-f003:**
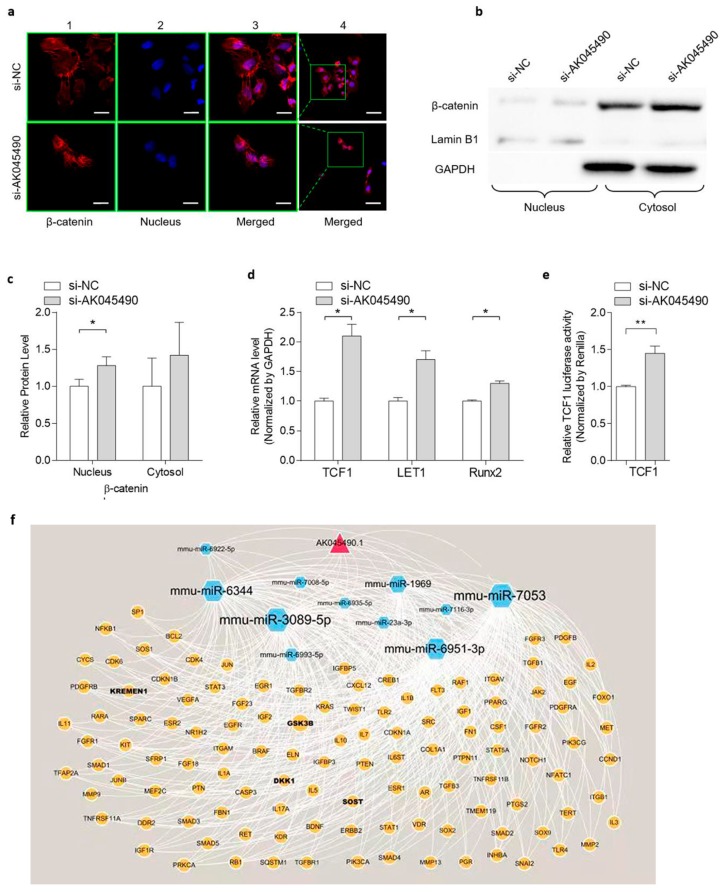
Knockdown of *AK045490* promoted β-catenin nuclear translocation and up-regulated the expression of TCF1, LEF1, and Runx2. (**a**) IHF staining showed the location of β-catenin in the cells transfected with *AK045490*-siRNA (si-*AK045490*) or scrambled-control-siRNA (si-NC), respectively. β-catenin was stained as red and nuclei were stained by DAPI showing blue. Bar (1, 2, 3): 20 µm. Bar (4): 50 µm. (**b**) Representative western blots of the nuclear translocation of β-catenin. The nuclear (nucleus) and cytosolic (cytosol) fractions of proteins isolated from *AK045490* knockdown cells and control cells were probed for β-catenin. Lamin B1 and GAPDH were used as internal controls for nuclear and cytosol fractions, respectively. Full unedited gels available in the Appendix A. (**c**) Quantification of nuclear and cytosol levels of β-catenin with Lamin B1 and GAPDH as internal control, respectively. (**d**) mRNA expression of TCF1, LEF1, and Runx2 as detected by real time PCR. (**e**) TCF1 activity in MC3T3-E1 cells transfected with *AK045490* siRNA, as detected by TOPflash luciferase reporter assay. (**f**) Pattern diagram showed the network of lncRNA–microRNA–RNA interaction. All data were expressed as mean ± SD. Student’s *t*-test was performed for comparison between two groups. *p* values less than 0.05 were considered significant in all cases (* *p* < 0.05, ** *p* < 0.01). *n* = 3 in each group.

**Figure 4 ijms-20-06229-f004:**
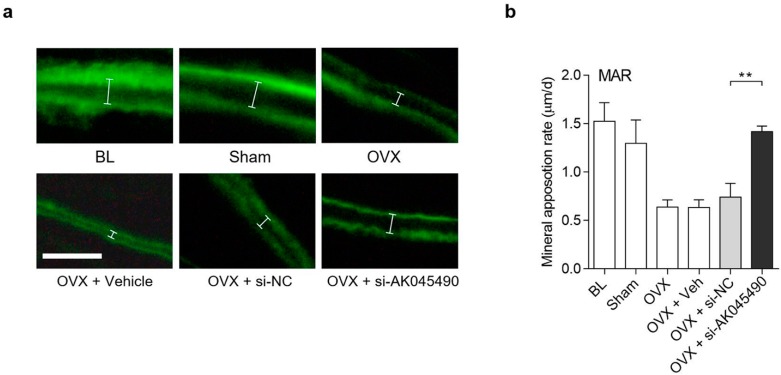
Promoting effect of AK045490 siRNA on calvarial bone formation in OVX mice. (**a**) Representative images showing calvarial mineral apposition rate of C57BL/6 mice after OVX treatment and siRNA transfection. White segments showed the width between the two lines. Scale bar = 20 µm. (**b**) Calvarial mineral apposition rates of C57BL/6 mice after OVX treatment and siRNA transfection. All data were expressed as mean ± SD. Student’s *t*-test was performed for comparison between two groups. *p* value less than 0.05 were considered significant in all cases (** *p* < 0.01). *n* = 5 in each group.

**Table 1 ijms-20-06229-t001:** Primer sequences for RT-PCR.

Target gene	Primer Direction	Sequence (5′–3′)
*AK045490*	Forward:	GCATTGTATCTCGCTCCACA
	Reverse:	TGTTGCCTACCTGCTTACTGC
*Alp*	Forward:	GTTGCCAAGCTGGGAAGAACAC
	Reverse:	CCCACCCCGCTATTCCAAAC
*Ocn*	Forward:	GAAGGCAACAGTCGATTCACC
	Reverse:	GACTGTCTTGCCCCAAGTTCC
*Col Ⅰα1*	Forward:	GAAGGCAACAGTCGATTCACC
	Reverse:	GACTGTCTTGCCCCAAGTTCC
*Tcf1*	Forward:	CAGAATCCACAGATACAGCA
	Reverse:	CAGCCTTTGAAATCTTCATC
*Let1*	Forward:	GATCCCCTTCAAGGACGAAG
	Reverse:	GGCTTGTCTGACCACCTCAT
*Runx2*	Forward:	CGCCCCTCCCTGAACTCT
	Reverse:	TGCCTGCCTGGGATCTGTA
*Gapdh*	Forward:	TGCACCACCAACTGCTTAG
	Reverse:	GGATGCAGGGATGATGTTC

*Alp*: alkaline phosphatase; *Ocn*: Osteocalcin; *Col Iα1*: collagen type I α1; *Tcf1*: transcription factor 7; *Lef1*: lymphoid enhancer binding factor 1; *Runx2*: runt-related transcription factor 2; *Gapdh*: glyceraldehyde 3-phosphate dehydrogenase.

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
