# Peer review of "Silencing of lncRNA AK045490 Promotes Osteoblast Differentiation and Bone Formation via β-Catenin/TCF1/Runx2 Signaling Axis"

_ijms, 2019, doi:10.3390/ijms20246229_

Round 1

Reviewer 1 Report

There is an unmet need of exploring the role of lncRNAs in bone formation. Li et al have the merit of undertaking this important exploration. They show that lncRNA AK045490 levels are higher in two mouse models of decreased bone formation: aging and ovariectomy. They claim that AK045490 may contribute to this decreased bone formation, because a treatment of the calvaria with AK045490 siRNA in the latter model leads to an increase in mineral appositional rate. By using MC3T3 cell cultures, they also show that AK045490 siRNA increases the expression of typical osteoblastic markers and transcription factors, and that this goes along with higher levels of catenin in the nucleus, as well as with increased bone nodule formation. These data deserve interest, but should be carefully presented and cautiously interpreted.

Specific comments:

Lines 123-128 are difficult to understand. The title of the paragraph states that knockdown of AK045490 promotes beta-catenin nuclear translocation (which I feel fits figure 3). However, the text line 126 states that knockdown of AK045490 inhibits the nuclear translocation of beta-catenin. Line 74 mentions AK016739, when the corresponding figure mentions AK045490 Would figure 2 not be more convincing if controls without any additive were also shown? Should figure 3f not be explained in greater detail? Figure 4: should the strong effect of the vehicle not be discussed? Some aspects of the discussion could be improved: avoiding to be merely a repetition of the results, or to be merely a review of a series of other reports. For example, the authors insist very much on clinical aspects – but do not mention whether anything is known about AK045490 in humans. It would be interesting to hear about this. Another example: are there drawbacks when using a MC3T3 cell line? These are transformed cells which may be regulated differently. Furthermore, an important contributor to osteoblastogenesis is cell proliferation which is difficult to study with transformed cells. Note the lncRNA Crnde affects mainly osteoblastogenesis through proliferation and this seems also mediated by catenin (Mulati et al 2020, Bone). In my opinion, the word “dramatic” is misused at various places of the text (for example lines 99, 230) Line 163, 240: is it correct to write postmenopausal mice? Line 148: delete (e), (f), (g) Line 328: it is not the cells but the nodules that are stained with alizarin Mistakes in English: throughout. Please check

Reviewer 2 Report

The paper identify a marker for osteoporosis with a direct involvement in osteoblastic differentiation. This is a basic study but with a potential high impact on the future management of osteoporosis, a frequent condition for postmenopausal women.

Found no methodological errors, but I consider that the manuscript should go through some revisions regarding English editing.

Author Response

The paper identify a marker for osteoporosis with a direct involvement in osteoblastic differentiation. This is a basic study but with a potential high impact on the future management of osteoporosis, a frequent condition for postmenopausal women.

Found no methodological errors, but I consider that the manuscript should go through some revisions regarding English editing.

Response: Thanks for the positive comments. The authors have revised the manuscript to address the Reviewer Comment by check the typographical errors sentence by sentence. All the revised information in the manuscript has been highlighted in yellow.

Round 2

Reviewer 1 Report

I feel the manuscript has improved. Still, I regret that the control without any additive is not shown in Fig 2. The authors refer to Fig 4 and recognize that the vehicle may have a toxic effect. Therefore, it is “healthy” that the reader is informed that that the investigation was performed under toxic conditions – and in my opinion it should be a general rule to include these data.
